# Developmental Changes in Patterns of Distribution of Fibronectin and Tenascin-C in the Chicken Cornea: Evidence for Distinct and Independent Functions during Corneal Development and Morphogenesis

**DOI:** 10.3390/ijms24043555

**Published:** 2023-02-10

**Authors:** Elena Koudouna, Robert D. Young, Andrew J. Quantock, James R. Ralphs

**Affiliations:** 1Structural Biophysics Group, School of Optometry & Vision Sciences, Cardiff University, Maindy Road, Cathays, Cardiff CF24 4HQ, UK; 2School of Biosciences, Cardiff University, Sir Martin Evans Building, Museum Avenue, Cardiff CF10 3AX, UK

**Keywords:** cornea, development, chick embryo, fibronectin, tenascin-C, keratocyte, neural crest

## Abstract

The cornea forms the tough and transparent anterior part of the eye and by accurate shaping forms the major refractive element for vision. Its largest component is the stroma, a dense collagenous connective tissue positioned between the epithelium and the endothelium. In chicken embryos, the stroma initially develops as the primary stroma secreted by the epithelium, which is then invaded by migratory neural crest cells. These cells secrete an organised multi-lamellar collagenous extracellular matrix (ECM), becoming keratocytes. Within individual lamellae, collagen fibrils are parallel and orientated approximately orthogonally in adjacent lamellae. In addition to collagens and associated small proteoglycans, the ECM contains the multifunctional adhesive glycoproteins fibronectin and tenascin-C. We show in embryonic chicken corneas that fibronectin is present but is essentially unstructured in the primary stroma before cell migration and develops as strands linking migrating cells as they enter, maintaining their relative positions as they populate the stroma. Fibronectin also becomes prominent in the epithelial basement membrane, from which fibronectin strings penetrate into the stromal lamellar ECM at right angles. These are present throughout embryonic development but are absent in adults. Stromal cells associate with the strings. Since the epithelial basement membrane is the anterior stromal boundary, strings may be used by stromal cells to determine their relative anterior–posterior positions. Tenascin-C is organised differently, initially as an amorphous layer above the endothelium and subsequently extending anteriorly and organising into a 3D mesh when the stromal cells arrive, enclosing them. It continues to shift anteriorly in development, disappearing posteriorly, and finally becoming prominent in Bowman’s layer beneath the epithelium. The similarity of tenascin-C and collagen organisation suggests that it may link cells to collagen, allowing cells to control and organise the developing ECM architecture. Fibronectin and tenascin-C have complementary roles in cell migration, with the former being adhesive and the latter being antiadhesive and able to displace cells from their adhesion to fibronectin. Thus, in addition to the potential for associations between cells and the ECM, the two could be involved in controlling migration and adhesion and subsequent keratocyte differentiation. Despite the similarities in structure and binding capabilities of the two glycoproteins and the fact that they occupy similar regions of the developing stroma, there is little colocalisation, demonstrating their distinctive roles.

## 1. Introduction

The cornea is the anterior transparent part of the eye. It is tough and accurately shaped optically, accounting for two-thirds of the eye’s refractive power. The cornea is a layered structure, with the thickest part being the stroma, which is found between the anterior epithelium and the posterior endothelium [1]. It makes up approximately 90% of corneal thickness and is a highly organised, dense connective tissue. In the human eye, it is composed of approximately 200 collagenous lamellae superimposed one upon another and each organised approximately orthogonally to its neighbours [2]. Keratocytes, the corneal stromal fibroblasts, are developmentally derived from neural crest cells that migrate into the corneal region and secrete the organised extracellular matrix (ECM). The ECM composition and architecture are crucial to the strength and transparency of the cornea.

The chicken cornea has the same fundamental organisation as the human cornea and is often used as a model system [1]. The development of the stroma starts when the optic vesicles extend from the brain to contact the cranial epithelium where the eye will form [3,4]. The epithelium secretes an orthogonally organised cell-free collagenous matrix beneath itself, the primary stroma, separating it from the underlying structures. An initial population of migratory neural crest cells enters the primary stoma and forms the corneal endothelium. As the primary stroma continues to be deposited, more neural crest cells enter the primary stroma and become presumptive keratocytes, which begin to secrete the definitive stromal ECM to form the secondary stroma [3,4,5]. Primary stroma formation stops during embryonic day (E)10, while neural crest migration and formation of new secondary stroma continue until E16 [6].

The adhesive glycoproteins fibronectin and tenascin are expressed in many developing connective tissues and have roles in the control of cell migration, adhesion and differentiation, as matrix-embedded signalling cues [7,8,9,10,11,12]. Fibronectin has a multidomain structure, with domains having multiple binding sites for cell surface receptors associated with adhesion and outside-in signalling, along with other ECM components. It forms a scaffold to which other extracellular proteins can bind and assemble [13,14,15]. The whole multidomain molecule occurs in two main forms—a globular, compact conformation and a fibrillar form, along with intermediates between the two, which give a “string of beads” conformation [15,16,17]. The transition between the two forms is cell mediated, occurring in association with integrins at the cell surface [15,18,19]. This can then potentially affect the assembly of other ECM components at the cell surface. Conformational changes in fibronectin can control biological functions, as binding sites for other molecules, including ECM, growth factors and cell surface receptors, are hidden or revealed in the different conformations [20,21]. Tenascin-C is similarly a multimodular and multifunctional glycoprotein that has structural and regulatory roles in tissue development, function and repair [22,23]. It binds to fibronectin, to other ECM components and to cell surfaces [24]. Cell binding to tenascin-C has various functions according to receptor and domain. Binding to different integrins can promote proliferation and inhibit apoptosis [24,25,26,27]. Binding to Toll-like receptor 4 promotes production of proinflammatory cytokines [28], and tenascin-C regulates binding of fibronectin to alpha5beta1 integrin by competing for the syndecan-4 receptor [29], conferring antiadhesive properties. 

Fibronectin is expressed by corneal fibroblasts [30,31]. It has been reported in developing cornea stroma, with expression reducing and eventually disappearing as the adult stroma forms [32,33,34]. It is rapidly re-expressed in pathology, injury and repair [8,9,10]. Corneal fibronectin forms string-like structures in the stroma that are associated with the epithelial basal lamina and extend into the stromal matrix [35,36]. There is some evidence that stromal fibroblasts/keratocytes interact with the strings, and it has been suggested that they provide cues for cells to position themselves in three dimensions during stromal development. Tenascin-C also occurs as a matrix component in the developing stroma, with expression reducing and eventually disappearing as the adult stroma forms [32,33,34]. It is also rapidly re-expressed in pathology, injury and repair [8,9,10].

Here, we describe at high temporal and spatial resolution the codistribution of fibronectin and tenascin-C in association with migrating neural crest cells and differentiating keratocytes. We show hitherto unreported and unexpected organisational features that suggest key controlling mechanisms for these matrix components in early development and subsequent morphogenesis of the corneal stroma.

## 2. Results

Fibronectin and tenascin-C are expressed at all developmental stages, exhibiting clear changes in arrangement and distribution in relation to cell migration and deposition as well as organisation of the secondary stroma.

### 2.1. Early Development: E6–E8

At early E7, the central region of the primary stroma contained no fibroblasts (Figure 1a–c) and had a uniform distribution of fibronectin, along with some focal regions of higher intensity with no clear substructure (Figure 1b,c). The fibronectin label was stronger in the stroma immediately above the corneal endothelium. Tenascin-C distribution was restricted to the posterior region of the primary stroma and the endothelium (Figure 1c). Its distribution was uniform, with no resolvable internal structure. At the periphery of the early E6 cornea, neural crest cells had invaded the primary stroma (Figure 1a,d), and linear fibrillar cords of fibronectin linked neighbouring cells together.

By late E6, the presumptive keratocytes had migrated throughout the corneal stroma (Figure 1e), interconnected with fibronectin cords. Tenascin-C distribution remained in the deeper regions of the stroma and had formed an orthogonal mesh of fibres (Figure 1f). Fibronectin cords were also present in the tenascin-C-rich area but did not co-organise with the tenascin-C mesh (Figure 1f). The tenascin-C mesh had developed further by E7 (Figure 1g), being more extensive but remaining in the posterior stroma. Even though localising to the same region of the corneal stroma as at E6, there was no structural similarity between the patterns of fibronectin and tenascin-C. By E8, additional fibronectin structures had developed in the anterior stroma (Figure 1h) as strings and bright, globular, focal spots of fibronectin, with the strings associated with the basement membrane of the epithelium and extending approximately 20–30 µm downwards into the stroma. High-resolution studies showed that strings and globular label were continuous, with globules forming beads along the linear strings (Figure 1i). Strings were often observed to spiral downwards into the stroma, with close associations with stromal cells (Figure 1j).

### 2.2. Middle to Late Development: E9–Adult

Fibronectin strings associated with the epithelial basement membrane persisted throughout development (Figure 2a–d). At E10, they extended downwards to deeper regions of the stroma where there was a deeper network of fibronectin cords between cells, as described above (Figure 2a). At E14 and E18, the strings were long—extending into the stroma between 50 and 150 µm; the full extent was difficult to determine, as it depends on the section plane (Figure 2b,d). At the junction between the strings and the basement membrane, strings were associated with the base of conical depressions of the basement membrane into the stroma (Figure 2c). The deeper network of cords present at E10 was absent from later stages. Fibronectin was present but was punctate rather than fibrillar in appearance (Figure 2b–d).

Tenascin-C distribution changed significantly over the mid to late developmental period. At E9, tenascin-C was a matrix component of the posterior region of the stroma, above the endothelial basement membrane (Figure 2e). By E10, tenascin-C was distributed more anteriorly, occupying the central region of the stroma, with low labelling intensity anteriorly and posteriorly (Figure 2f). By E14, this region was more anterior still (Figure 2g), although with a weakly labelled gap between it and Bowman’s layer (now visible by its positive tenascin-C label) until by E16, when it occupied the anterior third of the stroma and joined with the brighter labelled Bowman’s layer (Figure 2h). Tenascin-C was sometimes observed as string-like structures in the posterior stroma, distinct from the fibronectin strings, although this was not a consistent observation.

At all developmental stages beyond early E6, tenascin-C had a clear, approximately orthogonal mesh distribution of fibres in the ECM (see Figure 1f,g for early stages and Figure 2i,j for E10 and E18, respectively). In adult corneas, neither fibronectin nor tenascin-C were found in the corneal stroma, although fibronectin was present in association with the endothelial basement membrane (Figure 2k), and tenascin-C was present in Bowman’s layer (Figure 2l).

### 2.3. Fibronectin and Tenascin Interaction and Relationship with the Epithelial Basement Membrane

In stromal regions where both fibronectin and tenascin-C were concentrated, no clear association between their distributions could be identified at any stage in dual-label high-resolution studies (Figure 3a–h). During early development, where clear orthogonal networks of tenascin-C were visible, fibronectin as strands and globular, punctate foci were distributed amongst the tenascin-C mesh but showed no structural similarity with it (Figure 3a–d). At later stages (E16 onwards), where clear fibronectin strands linked with the epithelial basement membrane extended downwards into the tenascin-C lattice in the anterior stroma, there was no distributional correspondence between the two (E18, Figure 3e–h).

Dual-label studies investigating relationships between fibronectin strings and the basement membrane (Figure 3i–k) showed the presence of fibronectin in the strings and the basement membrane but that the key basement membrane components, type IV collagen (Figure 3i,j) and perlecan (Figure 3k), did not extend into the strings with the fibronectin. In the later stages, where tenascin-C was present in Bowman’s layer, fibronectin strings passed through the tenascin-C-rich layer en route from the basement membrane to the stroma, through pores in Bowman’s layer (Figure 3l).

### 2.4. Relationships of Fibronectin and Tenascin with Keratocytes

As shown above, in the early stages of cell invasion of the primary stroma, keratocytes were interconnected by fibronectin cords. Anteriorly, the cords formed a network that linked cells to one another and to the epithelial basement membrane (Figure 4a). In the later stages, where fibronectin strings were established, the strings were observed to be associated with keratocytes as they passed down through the stroma (Figure 4b–f). While the anterior ends of the strings were clearly associated with the epithelial basement membrane, there was no obvious termination point at the posterior end. Close associations with keratocyte nuclei could be observed, but no other associations were seen.

Conventional 3D projections from confocal datasets hinted at string–keratocyte associations (Figure 4b), but the use of Imaris 3D software enabled us to create rotating surface models that allowed retention of projection image sharpness in full rotation (Figure 4c–f). This revealed that fibronectin strings can pass through clusters of nuclei, weaving between them (Figure 4c), and with nuclei apparently wrapped partly around them (Figure 4d). This was indicative of a close association with cell surfaces, as cell outlines could not be visualised. Sometimes fibronectin strings had small side branches that appeared to be interacting with cells—they were positioned in intimate association with nuclei and, thus, presumably on the plasma membrane, between the nucleus, cytoplasm and fibronectin strings (Figure 4 e,f). Keratocytes were intimately associated with the tenascin-C mesh at all stages, essentially being tightly embedded within it. Nuclei could be observed enclosed in a dense mesh of tenascin-C fibrils (Figure 4g,h).

## 3. Discussion

This study shows that the developing corneal ECM contains complex, independent and changing arrangements of the matrix glycoproteins fibronectin and tenascin-C, both of which are important and sometimes interactive in function during cell migration and differentiation. They also interact with integrins for outside-in signalling at the cell surface. They have the potential to provide matrix-embedded cues for controlling neural crest cell migration into the developing cornea and for the positioning and differentiation of cells when they arrive.

The initial appearance of tenascin-C deep in the early primary stroma, prior to the migration of neural crest cells between the lens vesicle and epithelium, suggests that it is secreted by the endothelium and/or epithelium, which are in contact during early development. The antiadhesive function of tenascin-C has a role in separating opposing basement membranes to allow migrating cell access in craniofacial development [32,37,38], and has a potential role here in allowing the lens and corneal epithelia to separate as they are pushed apart by secretion of primary stroma, thus creating access for neural crest cells. Once the initial wave of migrating cells has formed the corneal endothelium [39,40,41], the tenascin layer may prevent later waves of cell migration from joining the endothelial layer, either directly or by preventing their interaction with fibronectin, ensuring that they populate the stroma.

As the secondary stroma becomes established with the initial migration of cells into the primary stroma, the tenascin-C mesh, similar to that of the developing collagen network [42], must be secreted by the presumptive keratocytes. Migrating neural crest cells secrete tenascin-C [43,44], which is consistent with this behaviour. Keratocytes regulate the rotating, branching and anastomosing collagen fibre macrostructure, and cell orientation relates to oriented collagen deposition [42,45,46], coincident with tenascin-C organisation. Tenascin-C is highly elastic [47] and could form an extensible link between cells and the collagen network, allowing cells to reorganise relative to existing collagen and deposit alternatively organised collagen. Tenascin-C associates with surface receptors, including integrins, syndecan-4 and Toll-like receptor 4, which each have distinct effects on cell behaviour [24], so feedback mechanisms can operate from tenascin-C back to the cell, perhaps to provide a matrix-embedded controlling mechanism for the deposition of oriented ECM. The relative movement of tenascin-C label from the rear to the front of the stroma is likely to relate to the way the primary stroma is deposited and then populated with migratory cells—the anterior stroma is younger, developmentally, than the posterior, and thus there could be a temporal relationship between tenascin-C expression and the state of differentiation of the presumptive keratocytes at a given developmental stage. The aspects of the distribution of tenascin we report in developing and adult corneas have been reported elsewhere, including in the epithelium and stroma of preterm and neonatal human corneas but not in adult stroma; in adults, the label was restricted to the corneal–scleral junction [34,48].

The control of keratocyte proliferation and phenotype is crucial to stromal development. Tenascin-C distribution indicates that it could control cell proliferation rates in the developing stroma. In early chick (E7) proliferation, rates are uniform across the stroma but later change to higher rates anteriorly than posteriorly (E11 onwards), matching the relative anterior shift of tenascin-C [49]. Tenascin-C promotes cell proliferation via alphaV beta3 integrin, previously reported in corneal stroma [24]. Spatial and temporal shifts in the expression of tenascin-C also occur in corneal wound healing [8,9,10]. Expression is initially in the posterior stroma, deep to the wound, and then moves anteriorly as healing proceeds before disappearing as repair completes. Tenascin-C is involved in phenotypic control of fibroblasts elsewhere—it acts with Twist1 transcription factor signalling to stably, but reversibly, activate fibroblasts in wound healing and permanently in fibrotic pathology and cancer [50]. Furthermore, Twist1 is expressed in corneal stroma and may be associated with stromal gene regulation [51].

Fibronectin cords between cells may organise and control cell migration into the periphery of the stroma. Migration is coordinated between cells so that they migrate as a sheet rather than as individuals [3,5,52], so the cords may keep cells together. It is unclear if the fibronectin cords are newly synthesised or a reorganisation of the primary stromal fibronectin, although in either case the formation of fibrillar fibronectin from secreted globular forms is cell mediated by interactions with integrins at the cell surface [11]. Interaction of cells with fibronectin allows outside-in signalling at the integrin-associated cell-matrix contacts, giving feedback to cells as they migrate and providing matrix-based cues with the potential for controlling migration, proliferation and differentiation.

Later prominent fibronectin strings, extending from the epithelial basement membrane into the stroma, have previously been reported [35,36]. In our specimens, they were present in all embryo stages, while they were not seen after E16 in previous results. The reasons are not clear, although possibly more modern second antibody labels and enhanced camera sensitivity may have contributed. Type IV collagen has been reported in the strings [35], a finding not replicated here, where we showed an abrupt boundary between the type IV collagen label characteristic of the basal lamina, between the basal lamina and strings; the basal lamina component perlecan similarly does not enter the strings. The reasons for this difference are unclear but could relate to differences in the antibodies used in the studies [53], although it has been shown that human and mouse corneas express a range of isoforms of type IV collagen and in mouse corneas, at least, expression is developmentally regulated [54,55]. Such studies have not been performed in chicken embryos, but similar expression could explain the different results with the antibodies used in this study and in the earlier study.

Fibronectin strings are clearly associated with the basal lamina of the epithelium by E8, and at this stage are the predominant form, as the intercellular cords are no longer present. A key question is, how do they form? The basement membrane is rich in fibronectin, and at the E8 stage, fibronectin can be detected in epithelial cells, which could readily explain basement membrane label but not strings extending down into the stroma. Therefore, they may be produced by the arriving presumptive keratocytes, which are clearly responsible for deposition of the secondary stroma components, with the intriguing observation that the strings run at approximate right angles to all of the other fibrous components deposited by these cells.

Once strings are formed, the close association of fibronectin strings with keratocyte nuclei is highly suggestive of cell–string interactions and thus acquisition of matrix cues by cells as they migrate into the stroma or indeed lay down the strings and secondary stroma. It has been speculated that strings could be used by cells for spacing within the stroma to control the width of the developing collagenous lamellae [35,36]. Formation of strings in association with the epithelial basal lamina could be important in the establishment of a positional system involving the developmentally oldest and most consistent part of the cornea, the corneal epithelium, with the basement membrane as a clearly defined boundary with the stroma. Interaction with strings can undoubtedly be the basis of a matrix-based signalling system to the cells—cells attach to fibronectin via a complex of integrins and syndecan-4 [13,56,57,58,59], and fibronectin can feed back mechanical signals through the adhesion complex [60,61]. Fibronectin fibrils are extensible under load, and as they extend, cryptic binding sites are revealed [20,21,62,63], which could alter the nature of the cell–string interaction and control cell behaviour as the cornea matures. The spiral nature of strings in the earlier embryos suggests a degree of extensibility as the cornea expands, and the observation that strings could be attached to the base of the cones of the basal lamina apparently pulled down into the anterior stroma suggests a firm attachment and perhaps a matrix-based mechanical signalling function. There is some reported variation across species with fibronectin distribution in adult corneas. The distribution observed here associated with the endothelium and Descemet’s membrane is consistent with reports of fibronectin mRNA expression in adult rat corneas [64] and fibronectin expression in adult rabbit corneas [65]; however, human corneas express fibronectin at the epithelial basal lamina [66].

Fibronectin and tenascin-C play major roles in tissue development, injury, disease and repair. Here, we show in unprecedented detail how these influential glycoproteins form sophisticated spatial and temporal relationships with cells and other key matrix components in the cornea as it develops. The corneal stromal cells secrete a complex, highly ordered ECM with fibronectin and tenascin embedded within it and show structural evidence of interaction with these matrix molecules that could be used in the control of position, proliferation and differentiation. This study forms a baseline for future detailed investigations into the roles of adhesive glycoproteins in corneal development, repair and tissue engineering. We recognise the requirement for a quantitative approach to understanding the control of expression of these molecules and their precise roles in corneal cell biology. Furthermore, we need to understand the distribution and roles of matrix receptors in migration and differentiation of presumptive keratocytes into the developing stroma.

## 4. Materials and Methods

### 4.1. Tissue Acquisition—Chick Embryos

Fertilised Delkolb chicken eggs (*Gallus domesticus;* Henry Stewart Ltd., Louth, UK) were incubated at 37 °C in a humidified incubator. Embryos were collected from embryonic day (E) E6 to E19. Adult chicken heads were also collected from freshly slaughtered animals from a local abattoir. All animal studies were carried out in accordance with the Association for Research in Vision and Ophthalmology statement and the UK Animals Scientific Procedures Act 1986.

### 4.2. Tissue Processing

Whole eyes (E6–8) or the anterior region of the eye containing the cornea (E9 onwards) were removed and fixed in 4% paraformaldehyde in phosphate-buffered saline (PBS, pH 7.4) for 30 min at room temperature. Specimens were washed thoroughly in fresh PBS and then infiltrated with 50% cryo-embedding compound (Bright Cryo-M-Bed, Bright Instruments, Huntingdon, UK) in PBS overnight at 4 °C before being transferred to fresh 100% OCT compound in a plastic mould and frozen at −85 °C. Cryosections were cut at 12 microns thickness using a Bright OTF 5000 cryostat, collected onto Superfrost Plus slides (VWR Ltd., Lutterworth, UK) and stored at −20 °C until required. A minimum of five eyes were used for each developmental stage. Three corneas were examined from adult chickens.

### 4.3. Immunohistochemistry

Cryosections were labelled by indirect immunofluorescence using single- and dual-label approaches. A minimum of five separate corneas were examined for each embryonic age studied. For single labels, sections were rehydrated with PBS containing 0.1% Tween20 (Sigma-Aldrich, Poole, UK; PBST) and incubated with primary antibodies overnight at 4 °C. These were the mouse monoclonals B3D6 to chicken fibronectin, M1B4 to chicken tenascin-C and 5C9 to chicken perlecan (all at 5 µg/mL in PBS) [67,68,69]. Sections were washed in PBST (three changes over 15 min) and then incubated with horse anti-mouse IgG conjugated with Dylight 488 secondary antibody for 6 h (Vector Laboratories, Peterborough, UK; 5 µg/mL). Sections were washed in PBST as above and mounted in Fluoroshield antifade mountant containing DAPI as a nuclear counterstain (Sigma-Aldrich, Poole, UK). Antibodies B3D6, M1B4 and 5C9 were obtained from the Developmental Studies Hybridoma Bank created by the NICHD of the NIH and maintained at The University of Iowa, Department of Biology, Iowa City, IA 52242, USA.

Dual labels for fibronectin and tenascin-C were possible due to the different IgG subtypes of the two primary antibodies—B3D6 is IgG2a and M1B4 is IgG1. Sections were incubated with a mixture of these two antibodies (5 µg/mL each) as above, washed and then incubated with a mixture of two isotype specific secondary antibodies: goat anti-mouse IgG2a-Alexa 488 conjugate and goat anti-mouse IgG1-Alexa 594 conjugate (5 µg/mL each; Invitrogen, ThermoFisher Scientific, Paisley, UK) and mounted as described above. Dual labels for fibronectin and type IV collagen used antibody B3D6 as above, mixed with polyclonal rabbit anti-chicken type IV collagen (5 µg/mL; anti-COL4A1 antibody, Antibodies-online, Aachen, Germany), using the same incubation times and procedures as described above. The secondary antibodies were horse anti-mouse IgG-Dylight 488 conjugate and horse anti-rabbit IgG-Dylight 594 conjugate (5 µg/mL each).

### 4.4. Fluorescence and Confocal Microscopy

Sections were examined using conventional epifluorescence widefield imaging using an Olympus BX61 microscope with a Hamamatsu ORCA Spark digital camera. High-resolution confocal laser scanning microscopy was performed using a Zeiss LSM 880 upright confocal microscope with an Airyscan Fast system for super resolution (max 140 nm lateral and 400 nm axial resolution). Image acquisition and 3D analysis were performed using Zeiss Zen Black software. BitPlane Imaris software was used to generate extended focus surface projections.

## Figures and Tables

**Figure 1 ijms-24-03555-f001:**
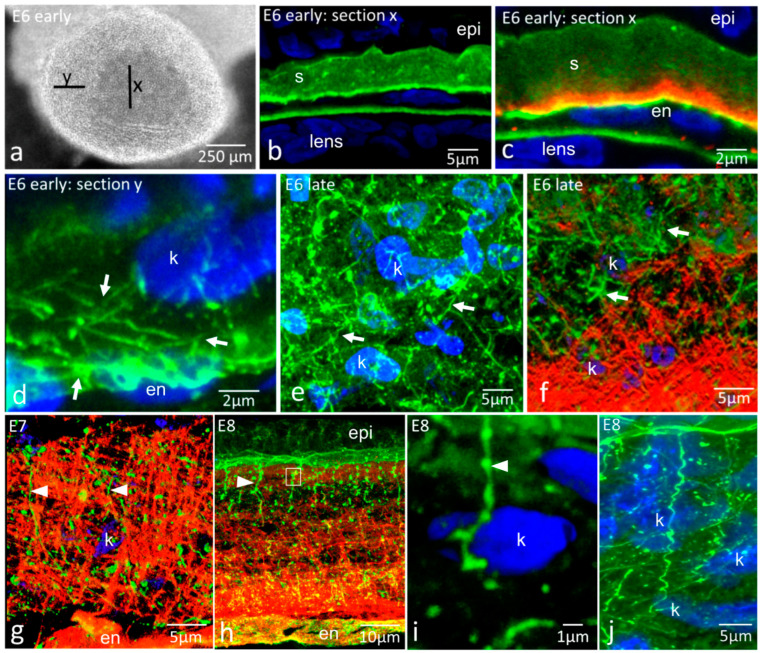
Immunohistochemistry for fibronectin (green) and tenascin-C (red) in early development and morphogenesis of the corneal stroma (E6–E8), demonstrating two distinct ECM glycoprotein networks. S, stroma; epi, epithelium; en, endothelium; k, keratocyte nucleus. (**a**) Whole-mount face-on view of early E6 cornea. Lines X and Y indicate the positions of corneal transverse sections shown in Figure 1b–d. Section X is central, and keratocytes have not yet populated the primary stroma—endothelial cells are seen beneath it. At section Y, keratocytes have invaded the primary stroma around the periphery of the cornea; (**b**–**i**) are transverse cryosections of embryonic corneas; (**b**) Early E6 central cornea, fibronectin label. Fibronectin is mostly uniformly distributed throughout the primary stroma, with slight enhancement at the endothelial side and some focal intensities. Note that presumptive keratocytes have not yet populated the central region. The lens capsule is also labelled; (**c**) Early E6 central cornea, dual immunolabel. Tenascin-C is present immediately above the endothelium. Colocalisation of fibronectin and tenascin-C (yellow) occurs in the endothelial basal lamina; (**d**) Early E6 peripheral cornea, fibronectin label. Presumptive keratocytes, as shown by their blue nuclei, are present within the peripheral stroma. Fibronectin cords run between keratocyte nuclei and between the nuclei and endothelial basal lamina within the stromal extracellular matrix; (**e**) Late E6 cornea, fibronectin label. Keratocytes are now present in the central region, having migrated throughout the stroma, and an extensive network of fibronectin cords is observed between them (arrows); (**f**) Late E6 cornea, dual label. Fibronectin cords (arrows) occur along with an orthogonal mesh of tenascin-C in the posterior part of the stroma. There is distributional overlap but no obvious structural colocalisation between the two glycoproteins; (**g**) E7 cornea, dual label. Tenascin-C appears highly ordered in the corneal stroma, intermingled with fibronectin (arrowheads), some of which is in the form of elongated strings associated with the epithelial basal lamina (left arrowhead). Fibronectin cords and foci run through the tenascin-C mesh and occur within the mesh but do not show structural colocalisation with it. This confocal projection was processed to emphasise tenascin and fibronectin organisation and has obscured some nuclear detail; (**h**) E8 cornea, dual label. Fibronectin strings (arrowheads) extend from the basement membrane of the epithelium downwards into the anterior region of the stroma, which is rich in highly ordered tenascin-C. Strings appear associated with numerous globular foci of fibronectin label (arrowhead). The box indicates the position examined in (**i**) E8 cornea, fibronectin label. Fibronectin strings (arrowhead) extending from the epithelial basal lamina into the stroma are often spiral in shape as they pass through the stroma and are very close to keratocytes. (**j**) E8 cornea, fibronectin label. Fibronectin Strings (arrowhead) extending from the epithelial basal lamina into the stroma, are often spiral in shape as they pass through the stroma and are very close to keratocytes.

**Figure 2 ijms-24-03555-f002:**
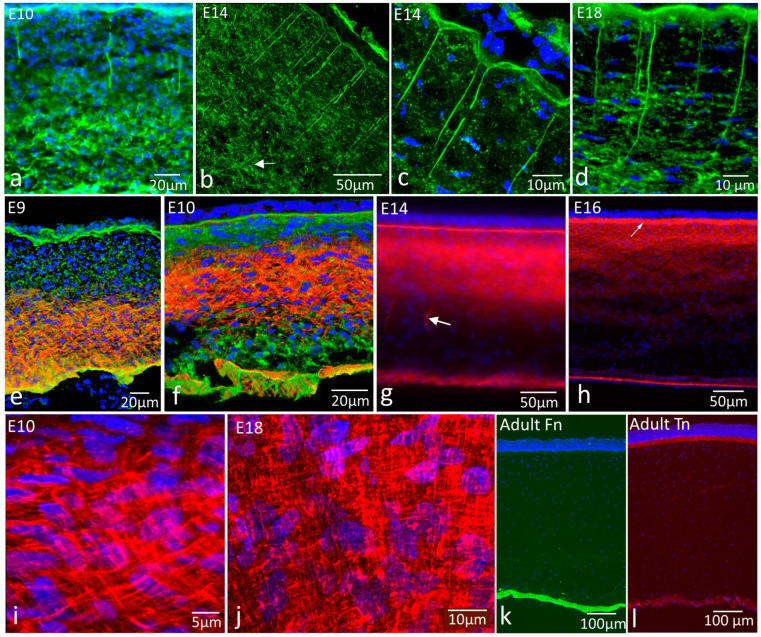
Immunohistochemistry for fibronectin (green) and tenascin-C (red) organisation in corneal development and morphogenesis. E9–Adult: distinctive glycoprotein networks persist, with some age-related changes, into late development but are lost in the adult cornea. Abbreviations: epi, epithelium; en, endothelium; Fn, fibronectin; Tn, tenascin-C. (**a**) E10 cornea, fibronectin label. Fibronectin strings (arrowheads) extend downwards from the epithelial basement membrane, while deeper in the stroma, there is a network of fibronectin cords (arrows) between keratocytes; (**b**) E14 cornea, fibronectin label. Numerous fibronectin strings (arrowheads) extend downwards into the cornea from their attachment points at the epithelial basement membrane. The fibronectin strings extend at least 100 µm, with some possibly further (arrow), although these could not be traced back to the basement membrane due to section plane; (**c**) E14 cornea, fibronectin label, higher magnification. Fibronectin strings (arrowheads) attach to the base of roughly conical depressions in the basement membrane. Strings were not always continuous, with gaps frequently being observed along them (arrow); (**d**) E18 cornea, fibronectin label. Fibronectin strings (arrowheads) were present, extending downwards through the anterior stroma and apparently ending in fibronectin-rich regions of the stroma (arrows) approximately 50–60 µm from the basement membrane; (**e**) E9 cornea, dual label. Tenascin-C occupies only the posterior region of the stroma, whereas fibronectin is distributed throughout; (**f**) E10 cornea, dual label. Tenascin-C is distributed more centrally, with regions of weaker label anteriorly and posteriorly. Fibronectin is distributed throughout the stroma; (**g**) E14 cornea, tenascin-C label. Tenascin-C occupies the anterior half of the stroma, with a weaker region of label between it and the strongly labelled Bowman’s layer. Tenascin-C is also present in occasional elongate string-like structures, distinct from the fibronectin strings, in deep regions of the stroma (arrow); (**h**) E16 cornea, tenascin-C label. Tenascin-C is strongly labelled in the anterior third of the corneal stroma and in Bowman’s layer (arrow); (**i**) E10 cornea, tenascin-C label, higher magnification, oblique section. Tenascin-C fibres are organised in a regular “orthogonal” mesh network; (**j**) E18 cornea, tenascin-C label, higher magnification, oblique section. Tenascin-C fibres remain as a regular “orthogonal” box network; (**k**) Adult cornea, fibronectin label. Fibronectin is associated with the endothelium, but little is detectable elsewhere; (**l**) Adult cornea, tenascin-C label. Tenascin-C labels Bowman’s layer and weakly in association with the endothelium.

**Figure 3 ijms-24-03555-f003:**
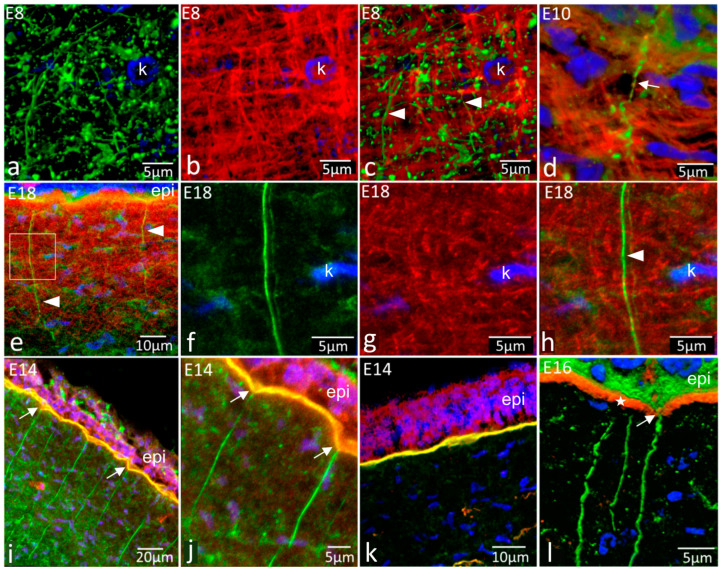
Immunohistochemistry for fibronectin (green) and tenascin-C (red) to show possible interactions and relationships with the epithelial basement membrane. K, keratocyte nuclei; epi, epithelium. (**a**–**c**) Dual labelling for fibronectin and tenascin-C, individual images ((**a**,**b**), respectively) and dual label overlay (**c**) of E8 cornea. Although present in the same regions of stroma, there is no clear codistribution—both have distinctive organisations, with fibronectin often in the spaces enclosed by tenascin-C fibrillar mesh; (**d**) Dual label for fibronectin and tenascin-C at E10. Fibronectin strings (arrow) pass into the tenascin-C-rich highly ordered matrix but show no structural similarity with it; (**e**) Fibronectin and tenascin-C at E18. Low magnification shows extensive tenascin-C labelling in the anterior cornea, with fibronectin strings (arrowheads) passing through it; (**f**–**h**) Dual-label images for fibronectin and tenascin-C. Individual images ((**f**,**g**), respectively) and dual label overlay (**h**) of E18 cornea at high magnification. No direct structural correspondence in the organisation of the two glycoproteins is evident. The fibronectin strings (arrowhead) pass through the highly organised tenascin-C network, which shows no structural disturbance associated with the fibronectin strings; (**i**,**j**) Dual label for fibronectin and type IV collagen (red) in E14 cornea, low and high magnification, respectively. Yellow/orange label indicates colocalisation in the epithelial basement membrane. The fibronectin strings do not contain any type IV collagen and have an abrupt boundary at their interface with it, at the base of the conical depressions (arrows); (**k**) Dual label for type IV collagen (green) and perlecan (red) in the E18 cornea. Perlecan colocalises with type IV collagen (yellow/orange signal) but does not extend from the basement membrane into the corneal stroma, so it is not a component of the fibronectin strings; (**l**) Dual label for fibronectin and tenascin-C in E16 anterior cornea. A fibronectin string is observed passing through a pore (arrow) in the tenascin-C-rich Bowman’s layer (asterisk) as it passes from the basement membrane into the stroma.

**Figure 4 ijms-24-03555-f004:**
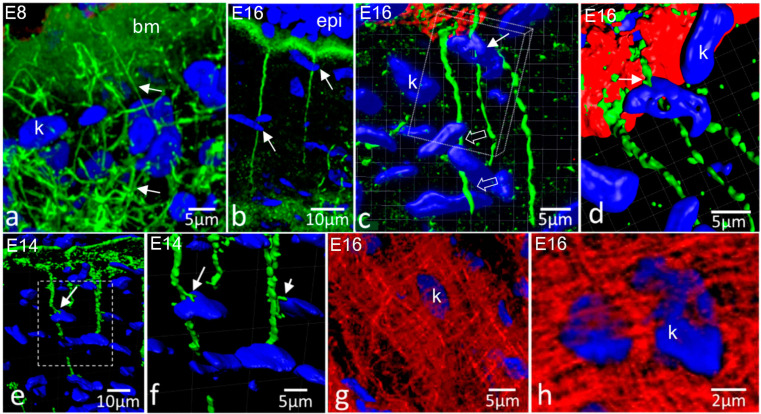
Fibronectin (green) and tenascin-C (red) interactions with keratocytes. bm, basement membrane; k, keratocyte nucleus; epi, epithelium. (**a**) E8 cornea, fibronectin label. 3D projection showing fibronectin cords/strings form links between keratocytes and between keratocytes and the epithelial basement membrane (bm; green sheet at upper part of image); (**b**) E16 cornea, fibronectin label. Fibronectin strings (arrowhead) pass into the stroma and run close to keratocytes, as shown by the presence of keratocyte nuclei near strings (arrows); (**c**) E16 cornea, dual label. Imaris 3D surface projection from dataset shown in (**b**), with tenascin-C channel added and dataset enlarged and tilted. Fibronectin strings pass very close to keratocyte nuclei (arrow) and, towards the bottom of the figure, a strand can be seen passing through a cluster of three keratocyte nuclei, running below the upper two and above the lowest (open arrows). The boxed volume is rotated and shown in (**d**); (**d**) Enlargement of the boxed region indicated in (**c**). The upper nucleus, as indicated by the white arrow in (**c**), is hooked around the fibronectin string (arrow). In addition, fibronectin strings can be observed emerging through the tenascin-C-rich Bowman’s layer (red); (**e**,**f**) E14 cornea, fibronectin label and Imaris 3D surface projections; (**f**) is an enlargement of the boxed region in (**e**). Fibronectin strings pass into the stroma from the basement membrane (top of image) into the stroma. Where keratocytes are encountered, as shown by their nuclei, small side branches of the fibronectin strings associate with them (arrows); (**g**,**h**) E16 cornea, en face section with tenascin-C label at medium and high magnification, respectively. Keratocytes, as shown by their nuclei, are closely embedded within the tenascin-C network.

## Data Availability

All image datasets, projections and 3D projections and videos used for this manuscript are available from the authors upon request.

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
