# Peer review of "Developmental Changes in Patterns of Distribution of Fibronectin and Tenascin-C in the Chicken Cornea: Evidence for Distinct and Independent Functions during Corneal Development and Morphogenesis"

_ijms, 2023, doi:10.3390/ijms24043555_

Round 1

Reviewer 1 Report

Reviewers Comments: “Developmental changes in patterns of distribution of Fibronectin and Tenascin C in the cornea: evidence for distinct and independent functions during corneal development and morphogenesis.”

Overall Comments

The paper’s data and observations match the title. The manuscript is well written with only minor spelling and grammar issues. However the figures all have errors that require careful attention. Overall the paper presents only qualitative assessment and superficial characterisation and could be much improved with even some basic quantification. The manuscript gives no information as to the number of samples used which is concerning as it is not clear if this is consistent with normal embryological development or a unique occurrence.

The authors mention in the introduction and again in the discussion – the interactions and signalling potential for tenascin C and fibronectin (TLRs, syndecan 4 and integrins). Given that this manuscript aims to characterise further the interaction between keratoytes and the aforementioned matrix proteins it seems a glaring omission that they have not included these receptors/adhesion molecules in their analysis.

The link between tenascin C and proliferation is interesting and could have been correlated with a proliferation marker to provide some evidence for this.

The authors comment that the presence of these fibronectin ‘strings’ has already been reported, as this is such a prominent feature of the paper, it makes it hard to see where and how much this manuscript advances the readership’s understanding.

I’m not sure I necessarily agree with their rational for conflicting evidence regarding collagen IV and perlecan associations with fibronectin (or lack thereof). If this discrepancy is due to technical issues I would expect more investigation before presenting this hypothesis, as it stands this work only confuses this observation.

Despite the very uniform nature of these strings penetrating directly and deeply through the stromal layers (unlike the projections shorter and more tortuous keratocyte) the authors do not seem to consider other cell types. While keratocytes are the primary cell in the stromal and known for matrix deposition, have the authors considered that the strings may be shaped by other cell types, neurons perhaps?

Figure 1

It would be a good idea to identify which markers are associated with which colours earlier on in the figure, either within the images themselves or at the start of the figure legend.

The authors state that “endothelial cells are seen beneath“ the corneal epithelium but have not provided any characterisation for these cells. This would help with identifying the orientation of the image, particularly in these very zoomed in images. Similarly for neural crest cells in the same figure.

Further explanation is needed to clarify the differences between images ‘b’ and ‘e’ as it is inferred both are central. Are both transverse sections or is ‘e’ en face? ‘Early’ and ‘late’ time points at the same day seem to be dramatically different if not. If ‘e’ is en face the authors should perhaps use a yz or xz slice to confirm that the cells sit within that layer and not above or below. Similarly with ‘g’ – additionally with ‘g’ perhaps showing the individual colours will help as the nuclear staining appears highly irregular.

It is unclear what the red scale bar in d is for and there are no arrows on the figure as the legend states. This should be addressed by the authors.

There is no figure 1j as referenced in the legend and in the results.

Figure 2

The authors state that tenascin C changes localisation as shown by 2e and f however it would be much more impactful if they could quantify this distribution over a number of sections and a number of corneas, even extending beyond the E9/E10 time points.

The fact that figure 2g uses green labelling for Tenascin C, particularly to show similar fibrillar structures to fibronectin is slightly confusing. Given that the authors only stain for 2 proteins in the whole paper, for consistency and ease of viewing I would recommend false-colouring the stainings accordingly.

Once again annotation is missing from this figure – there is no asterisk in ‘h’.

‘k’ and ‘l’ have labels “Adult Fn” and “Adult Tn”. These abbreviations are not explained in text and could be better conveyed.

Line 152: “Bowman’s layer” should this be “Descemet’s membrane”?

Figure 3

I think there is a mislabelling of this figure as the images do not seem to align with the legend or the text.

Once again annotation is missing from this figure.

Figure 4

Once again in this figure descriptions of “associations” with fibronectin and keratocytes are not validated by any form of analysis other than anecdotal observation and therefore are not convincing. Any way to quantify this would be extremely impactful. Even cytoplasmic/cell surface staining of the keratocyctes would help.

Some annotation is missing from this figure

Author Response

We are grateful to Reviewer 1 for their thorough review and helpful comments

Overall comments

Figure corrections and quantitation.  Figure corrections are discussed below.

Qualitative vs. Quantitative.  As the reviewer says, this is a qualitative, observational study rather than quantitative.  Quantitation can take many forms but we did not think it appropriate here.  At microscopical level, one could count strings, for example, or measure areas occupied by tenascin C label.  This would require acquisition of serial cryosections at all embryonic stages examined to ensure sampling across the corneas, a very difficult undertaking.  Area measurements are also dependent on having exactly the same orientation of transverse sections – sections are transverse but will rarely be at 90 degrees to the corneal surface, for technical reasons and also due to corneal curvature.  Other approaches could quantitate the proteins, eg by extraction and western blotting, or indeed by RNA extraction and QPCR, but this loses the regional significance of the distributions of these glycoproteins in the cornea. 

We have included the number of embryos examined – as the reviewer states this is fundamental and we are unsure how it was omitted – presumably an editing error on our part.

Cell interactions & signalling potential.  We agree that understanding the distribution of such receptors is important.  However there are many integrins that could be involved in Fn and TnC interactions, along with numerous Toll-like receptors (see manuscript discussion).  These receptors HAVE been reported in cornea, although not in a developmental context.  The amount of work necessary to follow these through development is a major project in its own right and beyond the scope of the work reported here.  Not discussing these would seem to be a significant omission from the manuscript. We have some preliminary results for certain integrins obtained after submission of the manuscript but these are not yet sufficiently conclusive to include in the manuscript.  

Tenascin C and proliferation.  We compare our distributions with a rigorous and high-quality study based on cell cycle analysis (using tritiated thymidine autoradiography) on chick embryo corneas.  This is excellent work even though it may be regarded as old and the distributions clearly relate to our findings. 

Fibronectin strings, previous reports.   It is of course true that the fibronectin strings have been reported previously, with fibronectin being described along with a range of other ECM components.  However our results add substantially to these findings.  Our studies are of much higher resolution and extend through a wider range of developmental stages, from very early through to adult.  We demonstrate at high resolution details of the basic architecture of individual strings at different developmental stages and provide some observational evidence of intimate association with stromal cells.  The interface with the fibronectin-rich basal lamina is clearly demonstrated, as is the later relation of fibronectin distribution and tenascin distribution at the epithelial basal lamina with fibronectin strings passing through the tenascin layer, as that layer is deposited after the fibronectin structures have formed.  This adds significantly to what was previously known about fibronectin organisation in corneal stroma.

Perlecan, collagen IV and fibronectin.  This section was a discussion of why there was a discrepancy between type IV collagen label in strings (in the early paper), which was not detectable in our study despite the fact that our collagen IV antibody was clearly working as expected (internal +ve control labelling of all basal laminas) and showed a very sharp boundary between basal lamina and fibronectin string.  The antibody used in the earlier study was not available and so we could not do direct comparisons.   It would be remiss of us not to comment on these findings, but perhaps could be regarded as inappropriate to speculate as why they might have occurred.  Accordingly we have modified the manuscript to leave this as unexplained. The use of perlecan was to see if another basal lamina component extended into the strings, which it clearly did not.   

Strings and other cell types.  The referee has a point – we do not comment on other cell types.  Nerves are visible in perlecan labelled material (not shown) but do not show any association with the fibronectin strings, which are present throughout, not just where the nerves run.  We have altered the text a little to clarify this.

Figure 1.  We are grateful to the reviewer for pointing these issues out and have made all of the corrections indicated.

Figure 2.  The reviewers comments were helpful and we have made the suggested changes.  However the use of “Bowmans layer” in line 152 is correct in the original manuscript – the tenascin rich region moves forwards and blends with Bowmans at the anterior of the stroma.

Figure 3.  We are grateful to the reviewer for pointing out these issues.  We accidentally put a draft version of the figure in the manuscript.  We have replaced this with the correct one and have checked the text and the legends against it.  We have made similar modifications to the above to help the reader.

Figure 4. As rightly pointed by the reviewer, annotations on the figure have been corrected. We indeed describe associations based on observation.  We agree that observations do not prove attachment to fibronectin strings, but clearly from the images and the scale bars there are sub-micron spacings between the fibronectin label and cell nuclei; spacing to the (invisible) cell membranes will be even closer.   

Thank you for your time.

Reviewer 2 Report

The authors wrote an interesting and scientifically solid article.

I'd add/increase the limitations section of the study

Author Response

We are pleased that the reviewer thought our work was interesting and scientifically solid.  At their suggestion we have added a brief limitations and future directions section to the final paragraph.

Thank you for your time.

Reviewer 3 Report

The ECM contains the multifunctional adhesive glycoproteins fibronectin and tenascin-C except collagens and associated small proteoglycans. Koudouna et al. reported that fibronectin and tenascin-C could be involved in controlling migration and adhesion and subsequent keratocyte differentiation. Moreover, there is little co-localization between them, demonstrating their distinctive roles. The manuscript is interesting. There are some minor defects.

1.      The authors used the chicken embryos as the model. Please add chicken in the title.

2.      Line 52. If the authors add references, it will be better.

Author Response

We are pleased that the reviewer found our manuscript interesting.  As suggested we have added “chicken” to the title and a reference for line 52.

Thank you for your time

Round 2

Reviewer 1 Report

This manuscript was initially submitted  with a number of proofing errors that could have been avoided with proper review by the authors. Whilst many of these have been corrected, it feels as though the authors have only sought to resolve the proofing errors and have declined to add any data that would enable the paper to progress the knowledge of the field.    
